# Molecular and Signaling Mechanisms for Docosahexaenoic Acid-Derived Neurodevelopment and Neuroprotection

**DOI:** 10.3390/ijms23094635

**Published:** 2022-04-22

**Authors:** Hee-Yong Kim, Bill X. Huang, Arthur A. Spector

**Affiliations:** Laboratory of Molecular Signaling, National Institute on Alcohol Abuse and Alcoholism, National Institutes of Health, Bethesda, MD 20892, USA; bhuang@mail.nih.gov (B.X.H.); spectora@mail.nih.gov (A.A.S.)

**Keywords:** docosahexaenoic acid, phosphatidylserine, *N*-docosahexaenoylphosphatidylethanolamine, *N*-docosahexaenoylethanolamine, synaptamide, synaptic membrane proteins, Akt, GPR110, ADGRF1, cAMP, PKA

## Abstract

The neurodevelopmental and neuroprotective actions of docosahexaenoic acid (DHA) are mediated by mechanisms involving membrane- and metabolite-related signal transduction. A key characteristic in the membrane-mediated action of DHA results from the stimulated synthesis of neuronal phosphatidylserine (PS). The resulting DHA-PS-rich membrane domains facilitate the translocation and activation of kinases such as Raf-1, protein kinase C (PKC), and Akt. The activation of these signaling pathways promotes neuronal development and survival. DHA is also metabolized in neural tissues to bioactive mediators. Neuroprotectin D1, a docosatriene synthesized by the lipoxygenase activity, has an anti-inflammatory property, and elovanoids formed from DHA elongation products exhibit antioxidant effects in the retina. Synaptamide, an endocannabinoid-like lipid mediator synthesized from DHA in the brain, promotes neurogenesis and synaptogenesis and exerts anti-inflammatory effects. It binds to the GAIN domain of the GPR110 (ADGRF1) receptor, triggers the cAMP/protein kinase A (PKA) signaling pathway, and activates the cAMP-response element binding protein (CREB). The DHA status in the brain influences not only the PS-dependent signal transduction but also the metabolite formation and expression of pre- and post-synaptic proteins that are downstream of the CREB and affect neurotransmission. The combined actions of these processes contribute to the neurodevelopmental and neuroprotective effects of DHA.

## 1. Introduction

Docosahexaenoic acid (DHA, 22:6n-3), the omega-3 polyunsaturated fatty acid that is highly enriched in the brain, is required for optimal neurodevelopment and function [1]. The maximum accumulation of DHA in the brain occurs late in pregnancy and the early postnatal period when brain growth, neurogenesis, and synaptogenesis are most active [2]. The important role of DHA in these vital developmental events is evident as DHA stimulates neurite growth and arborization, synapsin puncta formation, synaptic protein expression, and glutamatergic synaptic activity in primary embryonic hippocampal neuron cultures [3,4], while it improves the long-term potentiation (LTP), a cellular model of learning and memory, and increases the expression of synaptic proteins involved in neurotransmission in the mouse brain (Figure 1). DHA also promotes the differentiation of rat embryonic neural stem cells and stimulates hippocampal neurogenesis in the adult rats [5]. Furthermore, DHA enrichment in the brain results in an anti-apoptotic effect, promoting neuronal survival under adverse conditions [6]. These effects of DHA may contribute to the enhanced functional recovery following traumatic brain injury in mice after dietary omega-3 fatty acid supplementation [7,8] and the prevention of cognitive deficits in the offspring of maternal DHA-deficient mice by omega-3 fatty acid supplementation in the early perinatal period [9].

The beneficial actions of DHA may be due in part to a stimulation of phosphatidylserine (PS) production and the accumulation of DHA-containing PS (DHA-PS) in neuronal membranes, resulting in the modulation of signal transduction pathways vital for neuronal differentiation and survival [10,11,12]. DHA is also converted to lipid mediators that are neuroprotective [13,14] and activates signal transduction pathways that promote neurogenesis, neurite outgrowth, and synaptogenesis in developing neurons [15,16]. Taken together, these mechanisms appear to mediate the effects of DHA on neurodevelopment and neuroprotection (Figure 2).

## 2. Effects of DHA in Neuronal Membranes

DHA is taken up by the brain either as unesterified fatty acid or as DHA-lysophosphatidylcholine [18,19,20]. It is incorporated into membrane phospholipids, particularly the phosphatidylethanolamine (PE) and PS that are contained in the inner leaflet of the plasma membrane [1]. Enrichment with DHA can modulate the function of membrane proteins by interacting with these DHA- and/or PS-rich domains and influence downstream signaling [21,22]. The accumulation of DHA-PS in the inner leaflet of the neuronal plasma membrane facilitates signal transduction processes that are involved in neuroprotection [10,11,12].

### 2.1. Phosphatidylserine Synthesis

PS is synthesized by membrane-bound enzymes that catalyze serine base exchange reactions with phosphatidylcholine (PC) or PE [23]. The effect of DHA on PS synthesis has been investigated in neuronal cultures and microsomes prepared from the cerebral cortex [10,17,24]. These studies indicate that the PC and PE substrates containing *sn*-2 DHA are preferentially utilized for PS synthesis in the brain. Neuro 2A cultures incubated in media supplemented with DHA incorporate large amounts of DHA into PS, resulting in a doubling of the total PS content of the cells [10]. An increase in PS does not occur when the cells are incubated with DHA in serine-free media or in media supplemented with an equivalent amount of oleic acid, indicating the selectivity of this serine base exchange process for DHA. Consistent with these findings, PS synthesis in microsomes isolated from the cerebral cortex is higher from PC containing DHA compared to PC with other fatty acids in the *sn*-2 position [17]. Likewise, purified PS synthase 2, which has a key role in PS synthesis in the brain, utilizes PE containing *sn*-2 DHA more effectively than PE containing *sn*-2 oleate or arachidonate [24], contributing to the expansion of the PS pool in the brain after DHA supplementation.

### 2.2. Phosphatidylserine-Dependent Neuroprotective Signaling

A time-dependent increase in apoptotic cell death due to serum starvation is prevented when Neuro 2A cells are enriched with DHA [10]. This is associated with an increase in DHA incorporation into PS and the total PS content of the cells. The antiapoptotic effect does not occur when the cells are grown in a serine-free medium or enriched with oleic acid, indicating the DHA and PS dependence of this effect. The mechanism of this neuroprotective effect involves the membrane translocation of key kinases that is required for their activation. In response to growth factors or G-protein signaling, the Raf-1, Akt, or PKC is translocated to the membrane and this process is facilitated by increasing the membrane PS by DHA supplementation [10,11,12], as illustrated schematically in Figure 3. For example, increasing the plasma membrane PS in Neuro 2A cells facilitates the IGF-induced Akt activation, and decreasing the PS content using CHO mutant cells lacking PS synthase-1 exerts the opposite effects [25]. PS is also essential for the plasma membrane recruitment of 3-phosphoinositide-dependent kinase 1 (PDK1), one of the major upstream enzymes for Akt phosphorylation [26]. Through facilitating Akt signaling, DHA-enhanced PS enrichment in the plasma membrane plays a salvaging role in neuronal survival when cells are under adverse conditions [6,25].

#### Mechanism of Phosphatidylserine-Dependent Akt Activation

Akt is a kinase that inhibits pro-apoptotic proteins such as Bad and forkhead transcription factors and thus promotes cell survival [25]. Akt comprises three distinctive domains, including the *N*-terminal pleckstrin homology (PH) domain, the central kinase domain (KD) [27,28], and the C-terminal regulatory domain (RD). Akt activation requires two stepwise molecular events, membrane translocation and the conformational changes of Akt. The latter step enables Akt activation through phosphorylation at T308 of the kinase domain and at S473 of the regulatory domain by PDK1 [29,30] and the rictor-mTOR complex (mTORC2), respectively [31]. PS plays critical roles in both steps of Akt activation, as illustrated in Figure 4.

The translocation of cytosolic Akt to the plasma membrane is initiated by the interaction of the PH domain with membrane phosphatidylinositol 3,4,5-trisphosphate (PIP_3_), produced by phosphoinositide-3 (PI3) kinase upon growth factor receptor stimulation. Anionic PS, however, critically participates in the activation by securing the PIP_3_-triggered membrane translocation of Akt and facilitating its conformational changes for full activation. Through the interaction with positively charged residues of Akt at R15 and K20 in the PH domain outside the PIP_3_ binding pocket and at K419 and K420 in the regulatory domain [25], PS facilitates the interdomain conformational changes of Akt to expose T308 and S473 for phosphorylation by upstream protein kinases PDK1 and mTORC2, respectively. For the exposure of T308, in particular, both PIP_3_ and PS are required, so that PDK1 has access to phosphorylate Akt at T308 (Figure 4). PS also interacts with cationic residues R466 and K467 in the PH domain of PDK1. This PS–PDK1 interaction facilitates the association of PDK1 with the plasma membrane, enabling the phosphorylation of Akt at T308 [26]. As DHA enriches PS in the plasma membrane, the PS-dependent facilitation of Akt and PDK1 activation provides a mechanism for the neuroprotective effects of DHA by improving neuronal survival under adverse conditions. In contrast, increasing PS does not affect PI3 kinase activity [25].

## 3. DHA-Dependent Synaptic Membrane Protein Expression

The DHA status in the mouse brain affects the synaptic plasma membrane proteome, as demonstrated by nanoLC-ESI-mass spectrometry with ^16^O/^18^O labeling [32]. The comparative quantification of more than 200 synaptic membrane and membrane-associated proteins indicated that eighteen synaptic proteins were upregulated in the DHA-adequate brains compared to the DHA-deficient brains. Many of these, including munc18-1, PSD95, synaptic vesicle glycoprotein 2a/2b, synapsin 1a/b, contactin 2, bassoon, and glutamate receptors NR2B and AMPA2, have important functions in neurotransmission. Others, including synaptojanin-1, spectrin beta 2, contactin-associated protein 1, intercellular adhesion molecule 5, dynamin-1, PSD-93, synaptopodin, synaptotagmin I, and syntaxin1A are involved in synaptic vesicle trafficking and recycling processes. The protein network analysis indicated the involvement of the cAMP-response element binding protein (CREB) 1 and caspase-3 signal transduction pathways [32]. These findings suggest that the DHA-mediated expression of these synaptic plasma membrane proteins depends on the transcriptional activity of CREB1, while DHA mitigates the degradation of these synaptic proteins by suppressing caspase-3 activation.

## 4. Bioactive Metabolites Produced from Docosahexaenoic Acid

DHA is converted in the brain to bioactive metabolites that perform neuroprotective and neurotrophic actions. Docosanoids are formed by oxygenase pathways [33], and *N*-docosahexaenoylethanolamine (synaptamide) is formed by the *N*-acylation phosphodiesterase pathway [34,35]. In addition, DHA elongation products are converted in the retina to elovanoids that have antioxidant properties [14].

### 4.1. Neuroprotectins

DHA is converted to docosatrienes by the murine brain and human glial cells during ischemia-perfusion [33,36]. Neuroprotectin D1 (NPD1), the 10*R*,17*S*-docosatriene that is formed, inhibits leukocyte infiltration, IL-1β-induced NF-κB activation, and cyclooxygenase-2 expression and thereby has an anti-inflammatory effect [36]. NPD1 also promotes survival in cytokine-stressed human neural cells; suppresses the expression of the amyloid-β42-triggered activation of proinflammatory genes; and upregulates the anti-apoptotic genes *Bcl-2*, *Bcl-xl,* and *Bfl-1(A1)* [37]. In addition, the upregulation of cREL and the enhanced expression of BIRC3 may contribute to the neuroprotective effect [38]. Other docosatrienes have anti-inflammatory effects [13,39], and recent evidence suggests that the 10*S*,17*S*-dihydroxy-DHA stereoisomer of NPD1 also shows neuroprotective activity [40].

### 4.2. Elovanoids

Elovanoids (ELV) are oxidized metabolites synthesized from DHA elongation products containing 32- or 34-carbons [14]. The compounds formed are 20,27-dihydroxy-14Z,17Z,21E,23E,25Z,29Z-32:6 (ELV-N32) and 22,29-dihydroxy-16Z,19Z,23E,25E,27Z,31Z-34:6 (ELV-N34). These lipid mediators are produced in the retinal pigment epithelium by a series of elongation reactions mediated by the fatty acid elongation enzyme ELOVL4. They have antioxidant properties and prevent retinal degeneration by enhancing the expression of pro-survival proteins in cells subjected to uncompensated oxidative stress.

### 4.3. Synaptamide

Synaptamide is an endocannabinoid-like DHA metabolite present in mouse and rat brains [41,42,43]. It is synthesized in cultured mouse embryonic hippocampal neurons, mouse cortical neurons, rat neural stem cells, Neuro 2A cells, and embryonic hippocampal homogenates [35,41,44,45]. Studies in Neuro 2A cells indicate that DHA supplied as the unesterified fatty acid is utilized more effectively for synaptamide synthesis than DHA contained in lysophosphatidylcholine [35]. However, DHA from both sources is incorporated by Neuro 2A cells into *N*-docosahexaenoylphosphatidylethanolamine (NDPE), 80% of which is in plasmalogen form. *N*-Acylphosphatidylethanolamine (NAPE)-phospholipase D hydrolyzes this intermediate, releasing synaptamide into the extracellular fluid. Synaptamide synthesis in the bovine retina and brain also occurs by this *N*-acylation phosphodiesterase mechanism [34].

Synaptamide is hydrolyzed to DHA and ethanolamine by fatty acid amide hydrolase (FAAH) [16,46], and the amount of synaptamide recovered from neuronal cell and hippocampal homogenate incubations increases substantially if an FAAH inhibitor is present [41,44]. This suggests that much of the synaptamide that is produced in the brain is rapidly hydrolyzed after it is formed.

Synaptamide can be converted to oxidized metabolites that bind to the cannabinoid-2 receptor and have anti-inflammatory and tissue protective effects [42,43]. The 17-hydroxy metabolite is present in mouse brains [42], and the 19,20-epoxy derivative is synthesized by activated BV2 microglia [43].

#### 4.3.1. Actions of Synaptamide in the Nervous System

Synaptamide mediates the neurodevelopmental effects of DHA. The DHA-induced increase in neurite growth, synaptogenesis, and synaptic protein expression in mouse embryonic hippocampal neuron cultures is potentiated by the addition of the FAAH inhibitor URB597, and synaptamide produces these effects at much lower concentrations than DHA [41]. Inhibitors of lipoxygenase, cyclooxygenase, or cytochrome P450 do not affect the bioactivity of synaptamide, indicating that synaptamide itself is responsible for the potent bioactivity [41]. Similarly, comparatively low concentrations of synaptamide increase the expression of synapsins, the NR2B subunit of the glutamate receptor, and glutamatergic synaptic activity in the embryonic hippocampal neuron cultures. Synaptamide also mediates the DHA-induced neuronal differentiation of mouse neural stem cells [45]. This is associated with an increase in cAMP and the activation of the PKA/CREB signaling pathway. Likewise, an increase in cAMP mediates the synaptamide-dependent stimulation of axon growth in primary cortical neuron cultures [44].

The neuroprotective effects of synaptamide have been observed in experimental systems. Synaptamide inhibits lipopolysaccharide (LPS)-induced proinflammatory cytokine expression [47,48] and NF-κB RelA subunit translocation into the nucleus of BV2 and primary microglial cultures, and the intraperitoneal injection of synaptamide reduces LPS-stimulated neuroinflammation in mice [49]. Likewise, synaptamide stimulates axon growth in axotomized cortical neurons and retinal explant cultures, and the intravitreal injection of synaptamide promotes axon extension after optic nerve crush injury in mice [50]. The intraperitoneal injection of synaptamide immediately after repetitive mild traumatic brain injury also reduces optic tract gliosis, axon degeneration, and visual deficit in mice [51]. Similarly in rats, synaptamide decreases neuroinflammation, improves cognitive impairment after mild traumatic brain injury, and improves hippocampal neurogenesis after chronic constriction injury of the sciatic nerve [52,53]. Synaptamide mediates the beneficial effects of DHA on neurodevelopment and neuroprotection at nM concentrations [15]. This suggests the high potential of targeting synaptamide and its receptor for developing novel therapies for neurodevelopmental and neurodegenerative disorders.

#### 4.3.2. Role of GPR110 (ADGRF1) in Synaptamide Signaling

Synaptamide derived from DHA is an endogenous ligand of GPR110 (ADGRF1), an orphan receptor that belongs to the adhesion G-protein-coupled receptor (aGPCR) subfamily. The neurotrophic and neuroprotective effects of synaptamide, such as neurogenesis, neurite growth, synaptogenesis, and anti-inflammatory effects, are mediated by its binding to GPR110, activating the cAMP/PKA pathway and promoting PKC and CREB phosphorylation [15,49]. Synaptamide potently induces cAMP production with EC50 in the low nM range in cultured cortical neurons [15,50], neural stem cells [45] and microglia [51]. The disruption of this binding by GPR110 antibodies or GPR110 gene knockout abolishes the synaptamide-induced bioactivity. The central role of GPR110 in this process is illustrated schematically in Figure 5.

#### 4.3.3. Identification of GPR110 (ADRF1) as the Synaptamide Receptor

A pull-down strategy using a biotinylated synaptamide analogue which exhibits synaptamide-like bioactivity has been employed along with mass-spectrometric analysis to identify orphan GPR110 as the synaptamide receptor [15]. In living cells, GPR110 binds to bodipy-synaptamide, a fluorescent analogue of synaptamide that can be monitored microscopically, further confirming GPR110 as the target receptor of synaptamide. Blocking GPR110 by the *N*-terminal-targeting GPR110 antibody or GPR110 gene knockout abolishes the bioactivity of synaptamide, demonstrating that GPR110 is the functional receptor of synaptamide. The apparent Kd for the in vitro interaction of immunopurified GPR110 and synaptamide is estimated to be in the low nM range [15]. 

#### 4.3.4. Molecular Mechanism of GPR110 Activation by Synaptamide

Distinctively, aGPCRs contain a GPCR-autoproteolysis-inducing (GAIN) domain located at the *N*-terminal region immediately before the seven-transmembrane domain (7TM). Cleavage at the GPCR proteolytic site (GPS) in the GAIN domain results in the exposure of a conserved Stachel sequence. It is postulated that the Stachel sequence interacts with the 7TM and thereby activates the aGPCRs, but Stachel-dependent signaling can also occur in a manner that is independent of the autoproteolysis of the GAIN domain [54,55]. Interestingly, synaptamide binds to the GAIN domain and activates GPR110 signaling without self-cleavage at the GPS, and GPR110 is not activated by the Satchel peptide. These findings indicate that the mode of GPR110 activation is clearly different from other aGPCRs.

The molecular basis of the synaptamide-induced activation of GPR110 has been demonstrated by probing the conformational change of GPR110 in living cells using in-cell chemical cross-linking and quantitative high-resolution mass spectrometry [56]. Synaptamide induces conformational changes in the extracellular GAIN domain and the intracellular region involving TM6 and the *C*-terminal, where the receptor presumably interacts with G-protein or β-arrestin. According to the mutagenesis and functional assays combined with molecular modeling, a binding pocket is located in the interface between the two GAIN subdomains [56]. Synaptamide specifically binds to the pocket through the interactions of the polar ethanolamine head group; carbonyl group; and the hydrophobic DHA chain with GAIN domain residues Q511, N512, and Y513, respectively. The resulting conformational change in the intracellular region of GPR110 leads to G-protein activation and β-arrestin recruitment [56]. In short, synaptamide activates GPR110 through binding to the GAIN domain, causing an intracellular conformational change that initiates G-protein activation and downstream signaling, as illustrated in Figure 6.

## 5. Summary and Future Directions

Based on the molecular and cellular mechanisms for DHA-derived bioactivity, the strategies for neuroprotection can be devised as depicted in Figure 7. The enrichment of DHA in neuronal membranes increases the PS, which can facilitate the translocation and activation of key kinases such as Akt, Raf-1, and PKC and promote neuronal survival under adverse conditions. DHA enrichment also expands the precursor pool for the endogenous production of a variety of bioactive DHA metabolites such as synaptamide, providing a preventive strategy for accidental brain injury by improving spontaneous recovery. Synaptamide, in particular, by specifically binding to the GAIN domain of GPR110, activates cAMP signaling; promotes neurogenesis, neurite growth, and synaptogenesis; and suppresses neuroinflammation. It thus suggests the therapeutic potential of synaptamide analogues in neurodevelopmental disorders and neurodegenerative conditions as well as various central nervous system injury scenarios.

Further studies on the mechanisms that mediate the neurodevelopmental and neuroprotective actions of DHA are likely to provide new approaches for the prevention and treatment of neurodegenerative diseases. The possibility that neuronal signaling events in addition to Akt, PKC, and Raf-1 modulated by interaction with membrane lipids enriched with DHA should be explored. For example, the DHA-induced increase in PS may stimulate the intracellular transport of cholesterol in neurons in view of the recent finding that Aster protein binding to PS mediates cholesterol transfer from the plasma membrane to the endoplasmic reticulum [57,58]. In addition, more information is needed regarding the factors that modulate the production and function of the bioactive metabolites synthesized from DHA in the nervous system. The testing of these DHA-derived lipid mediators in animal models of brain injury and neurodegenerative disease are likely to suggest new therapeutic approaches with translational potential.

The finding that synaptamide mediates the beneficial effects of DHA in neurodevelopment and neuroprotection at nanomolar concentrations indicates its high therapeutic potential for developing novel therapies for neurotrauma, axonal injury, and neurodegenerative diseases. Because of its short half-life, the therapeutic potential is likely to depend on the development of analogues resistant to hydrolysis and oxidative degradation. A currently available prototype containing two methyl groups in the ethanolamine headgroup is resistant to hydrolysis, has a higher binding affinity to GPR110, and is more efficient and potent in triggering the cAMP production and the resulting beneficial neural regenerative actions [50] and anti-neuroinflammatory effects [51]. Further design based on the molecular structure of the binding pocket in the GPR110 GAIN domain should provide additional synaptamide analogues with improved therapeutic potential. In addition, further characterization of the structure and function of GPR110 and the identification of interacting proteins will provide new insight into the physiologic and pathologic involvement of this receptor.

## Figures and Tables

**Figure 1 ijms-23-04635-f001:**
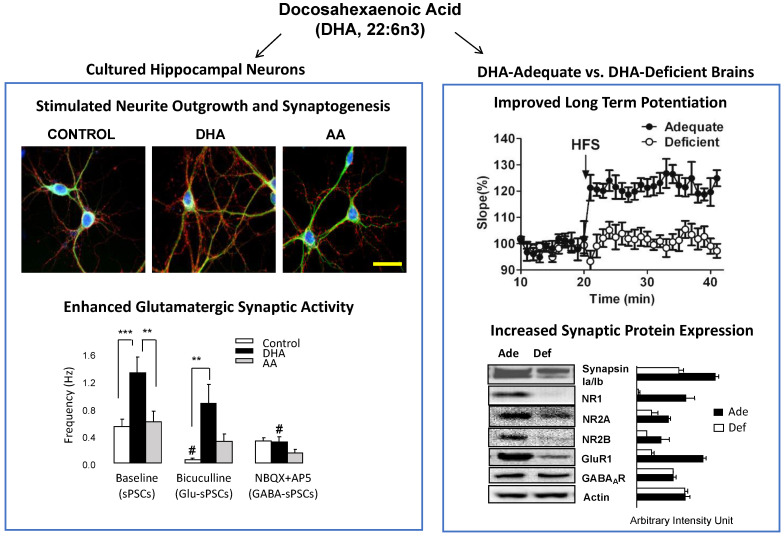
Neurodevelopmental effects of DHA. DHA among fatty acids uniquely promotes neurite growth, synaptogenesis, and glutamatergic synaptic activity in cultured hippocampal neurons. DHA-adequate mouse brains show improved LTP and increased synaptic protein expression compared to DHA-deficient brains. Markers used for the staining of hippocampal neurons include DAPI (for nuclei, blue), synapsin1 (a presynaptic marker protein, red), and MAP2 (a neuronal marker protein, green). AA, arachidonic acid. Adapted from reference [4].

**Figure 2 ijms-23-04635-f002:**
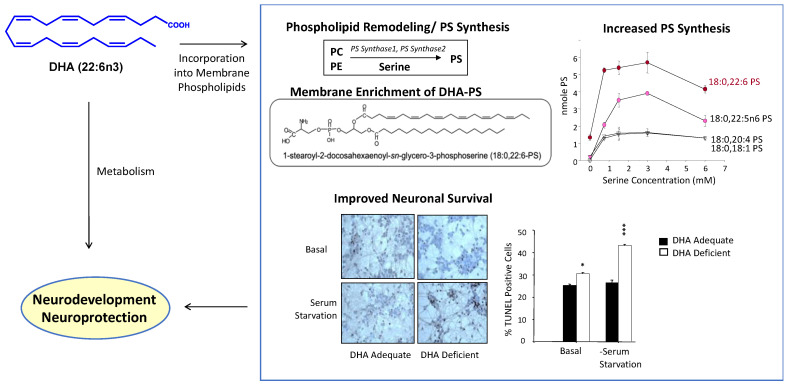
Neurodevelopmental and neuroprotective actions resulting from the membrane incorporation and metabolism of DHA. DHA increases PS in the neuronal membrane as DHA-containing phospholipids are the best substrate for PS biosynthetic activity. Susceptibility to neuronal cell death is reduced by increasing the PS level. Certain metabolites of free DHA can also exert neuroprotective effects. DHA, docosahexaenoic acid; PC, phosphatidylcholine; PE, phosphatidylethanolamine; PS, phsphatidylserine. Adapted from references [6,17].

**Figure 3 ijms-23-04635-f003:**
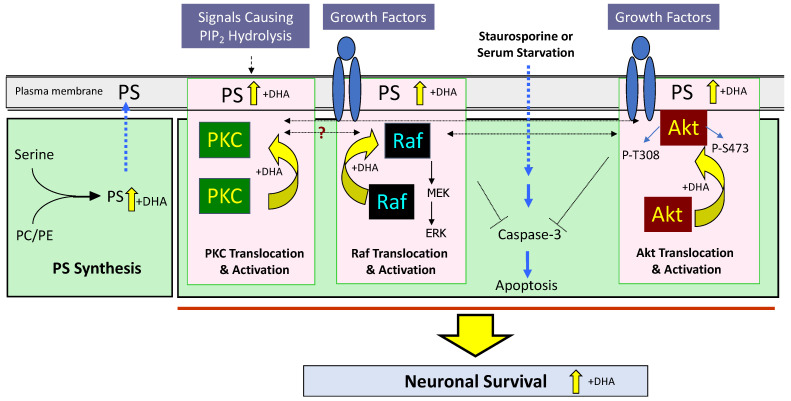
Membrane PS-dependent signaling mechanisms modulated by DHA that mediate neuronal survival under adverse conditions. By increasing PS in the neuronal membrane, DHA facilitates the translocation and activation of PKC, Raf-1, and Akt, positively influencing neuronal survival. PKC, protein kinase C; ERK, mitogen-activated kinase (MAPK); MEK, MAPK kinase. PC, phosphatidylcholine; PE, phosphatidylethanolamine; PS, phsphatidylserine. Adapted from reference [12].

**Figure 4 ijms-23-04635-f004:**
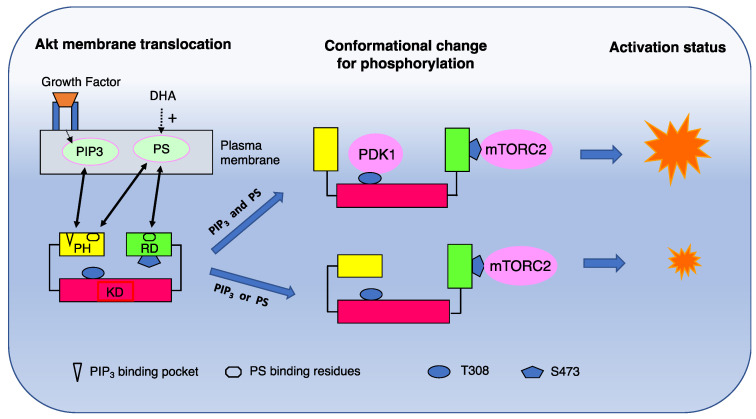
Molecular mechanism for PS-facilitated Akt activation. Upon stimulation of a growth factor receptor, cytosolic Akt is recruited to the plasma membrane though interacting with PIP_3_ and PS. The membrane interaction causes conformational changes of Akt, exposing T308 and S473 for phosphorylation by PDK1 and mTORC2, respectively. While S473 in the regulatory domain can be exposed by interacting with either PS or PIP_3_, both lipids are required for the conformational change of the PH domain to expose T308 in the kinase domain for full activation of Akt. DHA, docosahexaenoic acid; PS, phsphatidylserine; PH, pleckstrin homology domain; RD, regulatory domain; KD, kinase domain. Adapted from reference [25].

**Figure 5 ijms-23-04635-f005:**
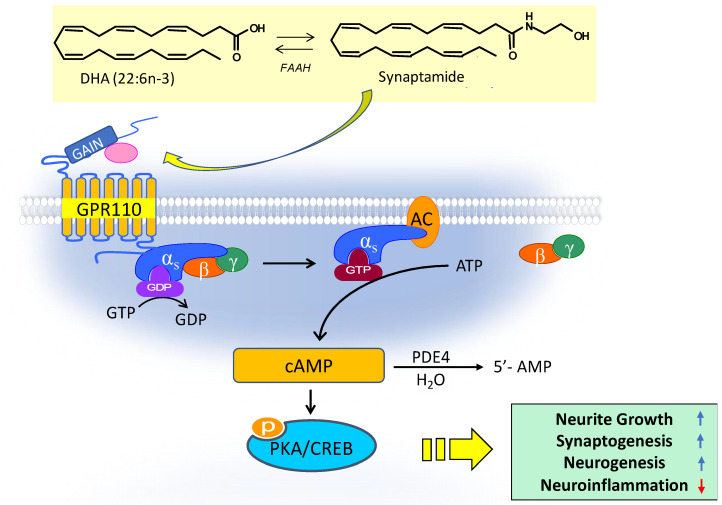
Synaptamide-activated GPR110 signaling. By binding to GPR110, synaptamide activates Gαs protein; increases cAMP; promotes neurogenesis, neurite growth, and synaptogenesis; and inhibits neuroinflammation. DHA, docosahexaenoic acid; FAAH, fatty acid amide hydrolase; GAIN, GPCR autoproteolysis domain; GTP, guanosine-5′-triphosphate; GDP, guanosine-5′-diphosphate; ATP, adenosine-5′-triphosphate; AC, adenylyl cyclase; CREB, cAMP response element-binding protein. Adapted from references [15,49].

**Figure 6 ijms-23-04635-f006:**
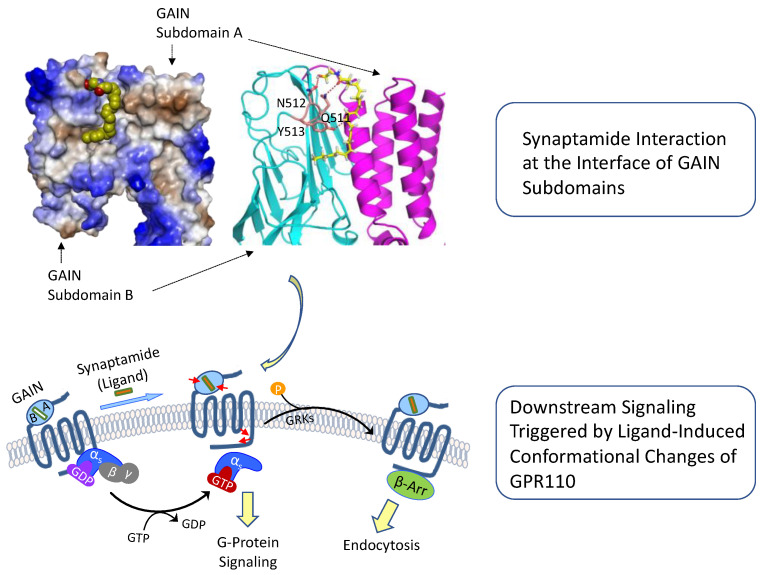
Activation of GPR110 and downstream signaling through synaptamide binding to the GAIN domain. Synaptamide is shown with a space-filling or stick-and-ball representation. Synaptamide binds to the interface between two subdomains, causing conformational changes in the GAIN and intracellular domains of GPR110 (depicted by red arrows) that trigger downstream signaling events. Adapted from reference [56]. GAIN, GPCR autoproteolysis domain; GTP, guanosine-5′-triphosphate; GDP, guanosine-5′-diphosphate.

**Figure 7 ijms-23-04635-f007:**
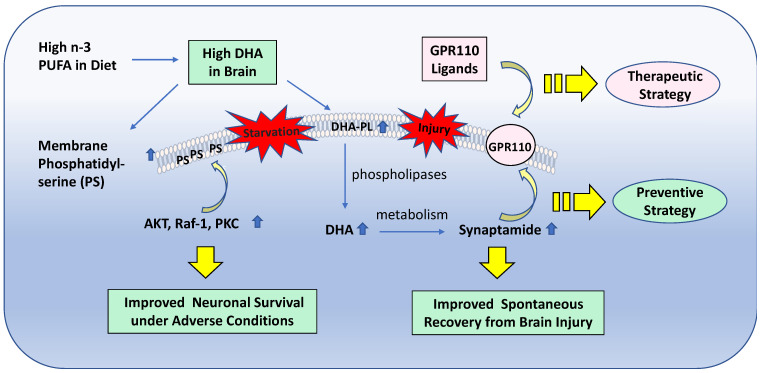
Preventive and therapeutic strategies for neuroprotection based on DHA-derived bioactivity. PUFA, polyunsaturated fatty acids; DHA, docosahexaenoic acid; DHA-PL, DHA-containing phospholipids. PKC, protein kinase C.

## Data Availability

Not applicable.

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
