# Peer review of "Molecular and Signaling Mechanisms for Docosahexaenoic Acid-Derived Neurodevelopment and Neuroprotection"

_ijms, 2022, doi:10.3390/ijms23094635_

Round 1

Reviewer 1 Report

This can be published except for minor spacing errors. Also line 51 and 107 in in Fig legends read referece not reference.

297 to 300 and refs are a different font. 

Author Response

Reviewer 1:

  1. This can be published except for minor spacing errors. Also line 51 and 107 in in Fig legends read referece not reference: Corrected.

  1. 297 to 300 and refs are a different font: Corrected.

Reviewer 2 Report

Kim and collaborators  concisely  review  the molecular and signaling mechanisms underlying the effects of DHA on neuronal development and its neuroprotective actions.

They specifically emphasize the effects of DHA on neuronal membranes, and the role of synaptamide, an endocannabinoid-like DHA metabolite, in the nervous system.

The mini-review is nice. but it would benefit from some minor revisions.

The Figure legends appear too brief.  More details need to be given. In addition, the reference from which the picture was adapted should be included in the legend

I suggest to carefully read the manuscript, as several typos need to be corrected

Line 99, deceasing  

Line 225 synapamide

….......

Author Response

Reviewer 2:

  1. The Figure legends appear too brief.  More details need to be given. In addition, the reference from which the picture was adapted should be included in the legend.

The figure legends have been revised as following.

Fig. 2:

DHA increases PS in the neuronal membrane as DHA-containing phospholipids are the best substrate for PS biosynthetic activity. Suceptibility to neuronal cell death is reduced by increasing the PS level. Certain metabolites of free DHA also can exert neuroprotective effects. 

Fig. 3:

By increasing PS in the neuronal membrane, DHA facilitates the translocation and activation of PKC, Raf-1 and Akt, positively influencing neuronal survival.

Fig. 4:

Upon stimulation of a growth factor receptor, cytosolic Akt is recruited to the plasma membrane though interacting with PIP3 and PS. The membrane interaction causes conformational changes of Akt, exposing T308 and S473 for phosphorylation by PDK1 and mTORC2 respectively. While S473 in the regulatory domain can be exposed by interating with either PS or PIP3, both lipids are required for the conformational change of the PH domain to expose T308 in the kinase domain for full activation of Akt.

Fig. 5:

Synaptamide-activated GPR110 signaling. By binding to GPR110, synaptamide activates Gs protein, increases cAMP, and promotes neurogenesis, neurite growth and synaptogenesis and inhibits neuoinflammation.  Adapted from reference 15 and 49.

Fig. 6:

Synaptamide binds to the interface between two subdomains, causing conformational changes in the GAIN and intracellular domains of GPR110 (depicted by red arrows) that trigger downstream signaling events.

Fig. 7:

Preventive and therapeutic strategies for neuroprotection based on DHA-derived bioactivity.

  1. I suggest to carefully read the manuscript, as several typos need to be corrected.

Line 99, deceasing:   Corrected.

Line 225 synapamide: Corrected.

Reviewer 3 Report

The present Ms by Kim HM et al explores and reviews several mechanisms that mediate the neurodevelopmental and neuroprotective actions of DHA and its derivatives. These compounds provide new and important issues for the prevention and treatment strategies against neurodegenerative diseases.

Their Ms is well designed and structured however in order to strengthen their study the authors should clearly explain some issues.

Main Remarks

  1. In the title, the authors should indicate the full term “DHA”.
  2. In Fig. 1 the authors should explain what are the used markers or proteins showed in red and green channels.
  3. The authors mentioned that “The later step enables Akt 112 activation through phosphorylation at T308 of the kinase domain and at S473 of the regu-113 latory domain by PDK1 [29,30] and rictor-mTOR complex (mTORC2), respectively

3.1. In this sense, the authors should explain and include the role of PI3K-mediated phosphorylation of AKT during neuroprotection induced by DHA.

3.2. In order to avoid a PI3K-mediated mechanism, they should revise the studies using inhibitor o similar method to block phosphorylation by PI3K.

3.3. The authors should provide more studies to explain DHA-mediated mechanism through mTORC2.

Author Response

Reviewer 3

  1. In the title, the authors should indicate the full term “DHA”. Corrected.
  2. In Fig. 1 the authors should explain what are the used markers or proteins showed in red and green channels.

The following sentence has been added in the legend.

Markers used for the staining of hippocampal neurons include DAPI (for nuclei, blue), synapsin1 (a presynaptic marker protein, red) and MAP2 (a neuronal marker protein, green).

  1. The authors mentioned that “The later step enables Akt activation through phosphorylation at T308 of the kinase domain and at S473 of the regulatory domain by PDK1 [29,30] and rictor-mTOR complex (mTORC2), respectively”

3.1. In this sense, the authors should explain and include the role of PI3K-mediated phosphorylation of AKT during neuroprotection induced by DHA.

We have added the following sentences.

Akt is a kinase that inhibits pro-apoptotic proteins such as Bad and forkhead transcription factors and thus promotes cell survival [25]. (Line 116-117). 

As DHA enriches PS in the plasma membrane, the PS-dependent facilitation of AKT and PDK1 activation provides a mechanism for neuroprotective effects of DHA by improving neuronal survival under adverse conditions. (Line 139-141)

3.2. In order to avoid a PI3K-mediated mechanism, they should revise the studies using inhibitor or similar method to block phosphorylation by PI3K.

We have previously demonstrated that the PS enrichment promotes Akt translocation and activation without altering the kinase activity of PI3K (ref: 25). We have added following sentence.

In contrast, increasing PS does not affect PI3 kinase activity [25]. (Line 141-142)

3.3. The authors should provide more studies to explain DHA-mediated mechanism through mTORC2.

DHA increases the PS level at the plasma membrane, promoting Akt membrane translocation and phosphorylation by PDK1 and mTORC2. This DHA-mediated mechanism has been discussed in the section 2.2. More studies detailing the DHA-mediated mechanism through mTORC2 have been described in reference 25.

Round 2

Reviewer 3 Report

Thanks for auhtors to correct the suggested changes.